# Trophic Transfer of Single-Walled Carbon Nanotubes at the Base of the Food Chain and Toxicological Response

**DOI:** 10.3390/nano12244363

**Published:** 2022-12-07

**Authors:** Majed Al-Shaeri, Lynn Paterson, Margret Stobie, Paul Cyphus, Mark G. J. Hartl

**Affiliations:** 1Centre for Marine Biodiversity and Biotechnology, Institute of Life and Earth Sciences, Heriot-Watt University, Edinburgh EH14 4AS, UK; 2Department of Biological Sciences, Faculty of Sciences, King Abdulaziz University, Jeddah 21589, Saudi Arabia; 3Institute of Biological Chemistry, Biophysics and Bioengineering, School of Engineering and Physical Sciences, Heriot-Watt University, Edinburgh EH14 4AS, UK

**Keywords:** ecotoxicology, nano-physiochemical properties, DNA damage, cytotoxicity, genotoxicity, metals interaction, trophic transfer

## Abstract

The potential for trophic transfer of single-walled carbon nanotubes (SWCNTs) was assessed using the green algae *Tetraselmis suecica* and the blue mussel *Mytilus edulis* in a series of laboratory experiments. Swanee River Natural Organic Matter (SRNOM)-dispersed SWCNTs were introduced into growing algal cultures. Light microscopical observations, confirmed by scanning electronic microscopy (SEM) and Raman spectroscopy, showed that SWCNT agglomerates adhered to the external algal cell walls and transmission electronic microscopy (TEM) results suggested internalization. A direct effect of SWCNT exposure on the algae was a significant decrease in growth, expressed as chlorophyll a concentration and cell viability. Mussels, fed with algae in the presence of SWCNTs, led to significantly increased pseudofaeces production, indicating selective feeding. Nevertheless, histological sections of the mussel digestive gland following exposure showed evidence of SWCNT-containing algae. Furthermore, DNA damage and oxidative stress biomarker responses in the mussel haemocytes and gill tissue were significantly altered from baseline values and were consistent with previously observed responses to SWCNT exposure. In conclusion, the observed SWCNT-algal interaction demonstrated the potential for SWCNT entrance at the base of the food chain, which may facilitate their trophic transfer with potential consequences for human exposure and health.

## 1. Introduction

The presence of carbon nanomaterials (NMs) in the environment is not novel [1]. Both carbon nanotubes (CNTs) and fullerenes (C_60_) have been discovered in 10,000 year old polar ice cores [2]. In the natural environment, they can be produced in hydrothermal vent systems [3,4] and during volcanic eruptions through processes such as Aitken mode nucleation, a consequence of simultaneous emission of substantive nuclei (e.g., nitric and sulfuric acids) [5]. However, it was not until the appearance of man-made NMs and nanoparticles (NPs), released into the environment either incidentally as the consequence of combustion, during production of engineered nanomaterials (ENMs) or other industrial, domestic and agricultural activities, that these materials have become an environmental concern [6,7,8]. Engineered single-walled carbon nanotubes (SWCNTs), first developed in 1991 [9], consist of a single layer of carbon atoms (graphene) rolled into a tube with a length of several μm and a diameter ranging between 0.4 and 3.0 nm. Their high aspect ratio, strength, light weight and electrical conductivity provide properties of great interest to industry, and consequently, SWCNTs find use in an ever-increasing number of products and applications, such as polymer composites, electrical and electronic components and displays, batteries, catalysts and many more. However, their increased production has led to increased incidences of engineered nanomaterials being released into the environment through a variety of routes during the product life cycle [10,11,12]. It is therefore more likely for SWCNTs to come into contact with marine algae, important primary producers at the base of the aquatic food chain [13,14,15,16]. Owing to their particular properties (aspect ratio, surface charge and area), single-walled carbon nanotubes (SWCNTs) have a strong ability to attract contaminants from environmental media [15]_,_ a property being increasingly used in nanomaterial-mediated remediation [11,17,18,19]. Phytoplankton is an important food source and feed additive in the commercial rearing of many aquatic organisms [18], and it has been shown that NPs can adhere to algal cell surfaces, restrict light accessibility to the algal cells, and thereby reduce photosynthesis, resulting in growth inhibition [20,21]. Questions remain regarding the fate, adherence and effect of SWCNTs during interaction with algal cells [22,23,24,25,26,27]. For instance, the agglomeration process of CNTs has been shown to be positively correlated with time-dependent algal growth inhibition, albeit at concentrations in excess of 25 mg L^−1^ [23,28]. Algae are also an important natural food source for filter-feeding bivalves, such as mussels. The feeding behaviour of mussels (*Mytilus* sp.) and other suspension-feeding bivalves has important water quality implications and has consequently been thoroughly studied in the absence and the presence of algal cells [29]. SWCNTs, both alone and in combination with other contaminants, have been shown to be toxic to aquatic organisms, inducing oxidative stress and DNA strand breaks in exposed mussels [10]. Therefore, the purpose of the present study was to assess the effect of SWCNT-contaminated algae on mussel feeding behaviour and the potential for the trophic transfer of SWCNT from primary producer to primary consumer. The hypotheses tested in the present study were that (i) mussels would continually filter algae in the presence of SWCNTs; (ii) mussels would ingest SWCNT-contaminated algae; (iii) mussels would produce loosely consolidated pseudofaeces, containing any freely dispersed SWCNTs, that would be injected back into the water column and quickly precipitate to the bottom of the tank [10]; (iv) SWCNTs will transfer from the ingested algae to the mussel; and (v) mussels will display a toxic response.

## 2. Experimental Section

### 2.1. Preparation and Characterization of SWCNT Suspension

The present paper reports findings from the second part of a larger study into the impact of SWCNTs, using the same batch of SWCNTs obtained from Merck Life Sciences UK Limited, Glasgow, Scotland (Cat No: 704121; manufacturer’s specifications: diameter 1.1 nm × length 0.5–100 μm). The SWCNTs were characterized (dispersion; dimensions; metal impurities) for the entire study, as described previously [10]. A SWCNTs stock (1 mg L^−1^) was prepared in distilled water using 0.02% Suwannee River Natural Organic Matter (SRNOM). Prior to use, the SWCNT stock was dispersed using an ultrasonic bath (Decon FS300 Frequency Sweep, Hove, UK) for two hours. SWCNT dispersion was assessed using SEM and Raman microspectroscopy was performed on the stock SWCNTs and in the tissue sections using an inVia Raman spectrometer with an integrated microscope (Renishaw, Wotton-under-Edge, UK), using a 785-nm laser operating at 5 mW output power. SWCNT agglomeration and surface charge under exposure conditions were assessed using a Malvern Nano-ZS Zetasizer (Malvern, UK).

### 2.2. Organisms

*Tetraselmis suecica* (strain PLY305) was obtained from the Plymouth Algal Culture Collection (The Marine Biological Association, Plymouth, UK). The algal cells were cultured in autoclaved 250 mL flasks containing 100 mL seawater to which 1 mL/L Guillard’s f/2 medium + vitamin mix 100 µL/L were added. The flasks were kept in a shaking incubator (Infors HT, Reigate, UK, Ref: 2011225), in which the shaking speed was 225 rpm and the temperature 23 ± 0.5 °C, with illumination by white incandescent lights (115 ± 15 μEm^−2^ s^−1^). Specimens of the blue mussel *Mytilus* sp. of similar length (5 cm ± 0.5) were collected from the walkway to Cramond Island at the mouth of the river Almond, in the north-west outskirts of Edinburgh, an area showing good water quality after recovering from historical discharges [30]. The mussels were transported straight to the laboratory and left for 48hrs to acclimatize in aerated filtered seawater (32 ppt/15 °C).

### 2.3. Interaction of Agglomerated SWCNTS with the Marine Green Algae, T. Suecica under Exposure Conditions

The *T. suecica* were exposed, in triplicate, to 5 µg L^−1^, 10 µg L^−1^, 50 µg L^−1^, 100 µg L^−1^ or 500 µg L^−1^ single-walled carbon nanotubes (SWCNTs) for 8 days. These concentrations were chosen based on preliminary dose response experiments with *T. suecica* and our previously published CNT toxicity data for *Mytilus* [10]. The highest concentration was used in the algae interaction experiments in order to establish a proof of concept, even though these are likely to exceed realistic environmental concentrations, for which there are currently no reliable PEC data available [8,31]. Examination of the SWCNTs on the algae *T. suecica* was performed using light microscopy (LM; Zeiss (Birmingham, UK) Axiophot microscope (5822/018030) equipped with a digital video camera using a ph₃-plan-Neofluar 100×/1.30 oil (∞/0.17) objective and an ocular lens Pl 10×/25), Raman mircospectroscopy (Renishaw - see below), scanning electron microscopy (SEM- see below) and transmission electron microscopy (Jeol JEM1400 Plus, TEM, Zaventem, Belgium - see below).

### 2.4. Raman Microspectroscopy

In order to identify the adsorbed and internalized SWCNTs in the algal culture medium, the algae were exposed to three replicates of SWCNTs 500 µg L^−1^ and control for 24 h in autoclaved 250 mL flasks. Following exposure, 5 µL of algae were placed on a quartz sample holder consisting of a vinyl spacer sandwiched between 2 quartz-cover slips. Raman microspectroscopy was performed on both the control and treatment samples using an in Via Raman spectrometer with an integrated microscope (Renishaw), using a 785-nm laser operating at 5 mW output power. A digital camera was used to capture transmitted optical images of the algal cells with a 0.75 NA Leica N-plan microscope objective (×50 magnification). The same procedure was carried out to identify SWCNT in mussel digestive gland tissue.

### 2.5. Scanning Electron Microscope (SEM)

A SEM was used to assess the adherence and agglomeration of the SWCNTs to the algal cell walls. Conventional methods for SEM analysis were followed [32,33]. In order to eliminate undesirable salt crystals, all samples were filtered through a pore size 0.45 µm filtration unit (Whatman membrane filters, cellulose nitrate; Cat No 7184 002; Maidstone, UK), and then the filters were rinsed with ultrapure water. Critical Point Drying was performed as follows: algal samples were slightly pre-fixed by immersing in a solution comprising 50:50 of acid Lugal’s iodine and 1.5% glutaraldehyde solution (sold as Grade I, 25% in H_2_O, specially purified for use as an electron microscopy fixative, from Sigma-Aldrich) and prepared in 0.1 cacodylic acid buffer (pH 7.3) and incubated at 4 °C overnight.

### 2.6. Transmission Electron Microscopy (TEM)

TEM was applied to determine the uptake of SWCNTs by the algae during exposure (500 µg L^−1^). The *T. suecica* culture samples were prepared for TEM analysis as follows. Algal culture samples were centrifuged at 474 g for 10 min. The centrifuged supernatants were discarded, and the pellets fixed in 4% glutaraldehyde and stored at 4 °C overnight. The samples were then washed 3 times in 0.1 M sodium cacodylate buffer (pH 7.24) for 10 min. The cells were post-fixed with 1% buffered osmium tetroxide and stored at 4 °C for 2 h. Prior to dehydration, the algal cells were washed again 3 times in 0.1 M sodium cacodylate buffer (pH 7.24) for 10 min. The fixed cells were dehydrated through a gradient series of acetone 35%, 50%, 75% and 95% for 15 min at each concentration, followed by 100% acetone [34,35].

### 2.7. Algal Growth Rates and Abundance

The effects of SWCNT-exposure on growth rates and abundance were estimated by measuring chlorophyll *a* concentrations as a proxy and calibrated using an improved Neubauer haemocytometer [34,36] Twenty four flasks were sterilized in an autoclave (Astell, Swiftlock Secure-Touch Touchscreen: Reference 2008129; Sidcup, UK), and then filled with 100 mL of cultured *T. suecica*. The treatments performed in triplicate were control, SRNOM and nominal SWCNT concentrations 5 µg L^−1^, 10 µg L^−1^, 50 µg L^−1^, 100 µg L^−1^, and 500 µg L^−1^, respectively. Sterilized tips were used for spiking throughout this experiment. The *T. suecica* were exposed for 8 days under environmentally-relevant growing conditions. The flasks were shaken (see point 2.2) to make sure no SWCNT could settle, and the algal cells in each flask were checked daily. The initial concentration of the algal cells was 54 cells µL^−1^. Chlorophyll *a* was measured as follows: 1ml algal samples were extracted to which 100 µL gum arabic+ 4ml acetone were added; all samples were kept in the dark at room temperature in order to extract the chlorophyll. Chlorophyll *a* was regularly extracted from t_0_ → t_3_ (0, 24, 48 and 72 h). In order to avoid the potential interference of SWCNTs with algal cells, the samples were filtered through 0.1 µm Omnipore hydrophilic PTFE polymer membrane filter (MerkMillipore Ltd., Cork, Ireland), and then measured using a chlorophyll fluorometer (Turner Design Instrument; model: #7200-00; San Jose, CA, USA) and compared against a reagent blank and expressed as µg L^−1^.

### 2.8. Cell Viability

Based on our previous concentration range experiments (see results section), the *T. suecica* were exposed to three replicates of SWCNTs 500 µg L^−1^ and controls without CNTs and incubated for 8 days. Both the control and the treated algae were filtered through a 100µm mesh immediately prior to analysis in order to eliminate any undesirable large residual particles. The cell viability was then determined using flow cytometry (CyFlow Reference 2005442-1711 from Partec, blue laser, 250 mW at the 424–488-nm band (FL1); green fluorescence (FL2) was detected at the 530 nm band; orange fluorescence (FL3) emitted by phycoerythrin (PE) at the 560 to 590 nm bands; chlorophyll (chl) which was detected as red fluorescence through a 660 to 700 nm band (FL4). In the flow cytometry, the algal cell size, density and intensity were assessed using forward scatter characteristics (FSC) and side scatter characteristics (SSC) plots to set up the gates to distinguish between the SWCNTs and algal cells [37]. This gating strategy can also be used to exclude debris as they tend to have lower forward scatter levels, which are often found at the bottom left corner of the FSC vs SSC density plot. Due to their smaller size, SWCNTs have smaller side scatter patterns compared to algal cells. Some of the algal cell samples were enhanced with SWCNTs before the flow cytometry analysis as a control to ensure that presence of SWCNTs within cells does not interfere with the laser detection system of flow cytometery [38,39,40]. All experiments were conducted in triplicate (*n* = 3).

### 2.9. Pseudofaecal Algal Cells and SWCNTs

As mentioned earlier, the aim of the present study was to assess and measure the effect of algal-SWCNT interactions on mussel feeding behaviour and health. This was conducted by determining the amount of pseudofaeces production, oxidative stress and DNA damage. The procedure for determining pseudofaeces production using flow cytometry was performed according to the previously published protocol [10]. The movement and feeding of the mussels viewed by naked eye and the pseudofaeces of the SWCNTs were recorded via video-camera recorder. Five starved mussels were added to triplicate glass tanks filled with 5 L of constantly aerated seawater (salinity range 33 ± 1 ppt; 15 °C), and then fed a known final concentration 622.67 cells µL^−1^ d^−1^ = 5500 mL^-1^ (FC) of (1) algal cells (*T. suecica*), (2) SWCNT 500 µg L^−1^ alone and (3) *T. suecica +* SWCNTs for 10 min. In order to observe any precipitating pseudofaeces, the mussels were transferred to clean seawater and left to depurate for 24 h. The control tank contained algal cultures *(T. suecica)* only, without mussels, to correct for algal cell division during the course of the experiment. At the end of the exposure-feeding period, the food (algae) and SWCNTs rejected by the mussels (pseudofaeces) were collected from the bottom of the tanks by pipet, immediately filtered through a 100µm mesh to eliminate large particles and the algal cells were analyzed using a flow cytometer (CyFlow Reference 2005442-1711 from Partec, Canterbury, UK, CT2 7FG.

### 2.10. Trophic Transfer of SWCNTs from Algae to Mussels

The uptake by mussels of algal cells (*T. suecica*) previously exposed to SWCNT (500 µg L^−1^), compared to an unexposed control culture, were assessed by measuring the ingestion rate of mussels during a 10 min exposure period (the clearance time in the experimental setup). The mussels were then dissected to extract their digestive glands and were assessed histologically for signs of algal and/or SWCNT ingestion and absorption. Histological tissue preparation was carried out as follows: the mussels were fixed in Davidson’s seawater fixative for 1 week. The fixative was decanted, and 70% ethanol added, for storage until processing [41]. The tissues were then processed using an automatic tissue processor (Shandon Duplex Tissue Processor; Shandon Diagnostics Ltd., Runcorn, UK) and embedded in paraffin blocks. The tissue blocks were sectioned using a rotary microtome (LKB, Bromma, 2218 Historange microtome, Ref. No. 577; Sollentuna, Sweden). All tissue sections were stained with haematoxylin and eosin. Finally, the slides were examined under a light microscope (Zeiss Axiophot microscope (5822/018030; Birmingham, UK) equipped with a digital camera using a ph₃-plan-Neofluar 40× objective). The Histological sections were also examined with Raman microspectroscopy to confirm the presence of SWCNT.

### 2.11. Toxicology

#### 2.11.1. Cell Isolation

The mussel haemocytes and gill cell suspensions were prepared according to the procedure for the clam *Tapes semidecusatus*, and adapted for mussels [42,43]. Briefly, valves were slightly wedged open to expose the posterior adductor muscle, and the water enclosed within it was drained. From each mussel, a volume of haemolymph was drawn into an equal volume of osmotically corrected Ca/Mg-free Hanks Buffered Saline Solution (HBSS + 22.2 g NaCl L^−1^ for 100% SW; this was diluted for lower salinities using an uncorrected medium; its osmolality was monitored using a Camlab freezing point depression osmometer) by inserting a 21-gauge needle on a 1 mL syringe into the posterior adductor muscle and transferred to an Eppendorf tube on ice or kept in the fridge overnight [44,45]. Following the extraction of the haemolymph, the gill single-cell suspensions were obtained by excising a gill; these were then transferred to a petri dish containing 2.5 mL of osmotically corrected HBSS and chopped 10 times using two fresh scalpel blades in a scissor-like movement. The HBSS containing tissue fragments was gently transferred to a 15 mL centrifuge tube, to which 2.5 mL trypsin (final conc. 0.05%) were added. The incubations were then gently rocked for 10 min at room temperature, after which 5 mL of HBSS was added and the suspension passed through a 40 µm sieve to remove any large fragments that remained. After centrifugation (800× *g* for 15 min), the supernatant was discarded, and the pellet was carefully resuspended in 0.5 mL of fresh osmotically corrected HBSS, and the tubes were placed on ice prior to further processing.

#### 2.11.2. Comet Assay

The Procedures for measuring DNA strand breaks in this study were carried out and adapted for mussels, as previously described [10,43,46]. The haemocytes were incorporated into the middle layer of a 1% (*w*/*v* PBS) agarose “sandwich” on frosted microscope slides. When the gels had set, the cells were lysed in a high-salt buffer [2.5 M NaCl, 10 mM Tris, 100 mM EDTA, 1% (*v*/*v*) Triton X-100, and 10% (*v*/*v*) DMSO, pH 10.0] for at least 90 min at 4 °C in the dark. Following lysis, the slides were gently placed into a horizontal electrophoresis tank and covered with an alkaline solution (0.3 M NaOH, 1 mM EDTA, pH 10) for 30 min at 4 °C in the dark to allow the DNA to unwind [47,48]. Without changing the electrolysis solution, a 25 V cm^−1^, 300 mA current was applied for 25 min and then neutralized three times with Tris buffer (0.4 M Tris-HCl, pH 7.4) at 5 min intervals. The slides were then washed with distilled water and stained using two to three drops of Gelred^©^ (0.02% in distilled water) for 5 min. Finally, the slides were again washed with chilled distilled water, cover slips were placed over the gels, and 50 randomly chosen nucloids per slide were analyzed using a Zeiss Axiophot epifluorescence microscope equipped with a Zeiss AxioCam MRm digital camera (400× magnification; Birmingham, UK), and DNA damage determined using the image analysis software package Comet Assay IV (Perceptive Instruments) and expressed as % tail DNA.

#### 2.11.3. Oxidative Stress

The oxidative stress indicators, expressed as superoxide dismutase (SOD) activity and lipid peroxidation, were measured in the gill tissue homogenates, as previously described [10]. In order to prepare the tissues for the oxidative stress assays, part of the gills excised during gill cell isolation for the Comet assay were diverted and immediately immersed in liquid nitrogen and stored at (−80 °C) until further use. The tissues were then homogenized on ice with 1:5 volumes of buffer (Tris–HCl 50 mM, 0.15 M KCl, pH 7.4), and centrifuged at 10,000× *g* for 20 min at 4 °C. The pellet was retained for thiobarbituric acid reactive substance (TBARS) analysis, whilst the supernatant fraction was centrifuged at 40,000× *g* for 60 min at 4 °C, in order to obtain the cytosolic fraction, which was then used to analyze the Superoxide Dismutase (SOD) activity. TBARS where determined according to the procedure described for trout [49]. To each well of a 96-well microtiterplate 40 µL of the gill homogenate, 10 µL of BHT, 140 µL of PBS, 50 µL of TCA and 75 µL of TBA were added. The plates were then incubated for 60 min at 50 °C, read at 530 nm and 630 nm (correction for turbidity: 530 nm–630 nm) against a TEP standard series (0.5, 2.5, 5, 15, 25 nM) and expressed as nmol mg^−1^ protein. The total homogenate protein was determined according to Bradford [48]. The SOD was determined using a kit (SIGMA 19160).

#### 2.11.4. Data Analysis

Prior to analysis, the Comet assay slides were coded to avoid operator bias. Following arcsin transformation, the data were statistically analyzed using a one-way ANOVA, followed by a Tukey all pairwise multiple comparison procedure; a *p* < 0.05 was considered significant [50,51]. The oxidative stress data were tested for normal distribution and statistically analyzed using a one-way ANOVA, followed by a Tukey as described above.

## 3. Results and Discussions

The spectrophotometric analysis demonstrated the effect of sonication on improving the dispersion of the SWCNT in 0.02% Suwannee River natural organic matter (SRNOM), and further SEM analysis showed that the SWCNTs formed loose agglomerates, dispersed throughout the stock suspensions. The SEM micrographs (Figure 1A) show the matrix of the SWCNTs dispersed in 0.02% SRNOM, which may possibly coat the SWCNTs [43], therefore increasing the thickness of what appear to be individual SWCNTs (Figure 1B). The zeta potential, observed under exposure conditions in slightly alkaline seawater, was −8.34 to −15.93, indicating a negative surface charge (Table 1). To the naked eye, the SWCNTs seemed to disperse well with SRNOM in seawater. DLS analysis, previously reported by us [10] and others [52,53], showed that the SRNOM-dispersed SWCNTs remained longer in suspension than those without SRNOM, and eventually became agglomerated in a concentration-dependent manner. Furthermore, the data showed that under stable pH conditions (pH 8), increases in the size of the resulting SWCNT agglomerates were also concentration-dependent (Table 1).

### 3.1. SWCNT Interaction with Algae

Using light microscopy and SEM, the algal cells grown in seawater and spiked with 500 µg L^−1^ SWCNTs were observed to sequester the SWCNTs on their external cell walls (Figure 2 and Figure 3). The Raman spectrum associated with the light micrographs clearly showed peaks at (RBM) 268 cm^−1^, with the D band at 1290 cm^−1^, G band at 1590 cm^−1^, and G′ band at 2585 cm^−1^, characteristic of 1.1 to 1.5 nm diameter SWCNTs [54,55] (Figure 4). Long et al. [55] showed that algal cells exposed to multi-walled carbon nanotubes (MWCNT) became largely entangled with MWCNT agglomerates and displayed no sign of mitotic divisions. They further observed that most of the algal cells trapped in the MWCNT layer lost their cellular integrity, with cytoplasm outflow, indicating irreversible cell damage. Schwab et al. [29] reported a shading affect attributed to the agglomeration of both SWCNTs and MWCNTs, leading to a reduced algal growth rate. Similar observations were made in the present study, where, at the end of the exposure to ≥100 µg L^−1^ SWCNTs, most of the algal cells were visibly entrapped within the SWCNT agglomerates (Figure 2A,B). This caused a decrease in chlorophyll *a* concentrations and decrease of up to 87% in viable algal cells, (Figure 5 and Figure 6), and a concomitant inhibition of algal growth was observed (Figure 7). This could be reduced by optimizing the shaking intensity during incubation [55,56,57]. Long et al. [55] reported that increasing the shaking speed may increase the possibility of a physical interaction between multi-walled carbon nanotubes (MWCNTs) and algal cells, and thus the toxicity of MWCNTs, but that the effect (i.e., growth inhibition) remained largely unaffected by shaking speeds >180 rpm. A similar result has been shown in the present study with SWCNTs, using a shaking speed of 225 rpm and light intensity of 115 ± 15 μEm^−2^ s^−1^. However, others have shown that shaking speeds ≥180 rpm could actually have a stimulating growth effect on the algae, caused by the increased light availability [58]. The shading hypothesis afforded by CNTs adhering to external algal surfaces [22,30] and associated reduced cell viability, growth rate and chlorophyll *a* concentrations in exposed phototrophic biota [23,37,52,59] is consistent with the fact that CNTs are relatively opaque. The physical interaction between the SWCNTs and *T. suecica* was directly observed here using LM, Raman spectroscopy, SEM, and TEM. The SWCNTs attached and adhered above, below and in between the algal flagella, diminishing the flagellar motility, and thus reducing the mobility of the algal cells. The Raman spectroscopy confirmed the presence of SWCNTs on the algae, and it was thus concluded that the detrimental effects on algal growth rates were the result of the cells being smothered by agglomerations of SWCNTs on the cells (Figure 3), thus reducing the residence time of the algae in suspension. Youn et al. [30] exposed freshwater green algae (*Pseudokirchneriella subcapitata)* to 500 µg L^−1^ SWCNTs dispersed in gum arabic (GA). Comparing the presence of SWCNT in *P*. *subcapitata* [30] and *T. suecica* (present study) showed that the different dispersants involved (1% GA and 0.02% SRNOM) did not affect the Raman spectroscopic identification of the SWCNTs. Furthermore, the Raman spectroscopy data showed that the SWCNTs were similar in surfactant coverage and agglomerate state throughout algal growth [29]. SEM images, in the present study, showed that the appearance of the algal cell shape changed following exposure to the SWCNTs compared to the controls (Figure 2A,B). The ≥100 µg L^−1^ SWCNT agglomerates were shown very clearly to surround the algal cells. This observation is consistent with previous accounts of the shading effects of agglomerations of SWCNTs and subsequent algal growth inhibition [23,52,59]. Some life forms, such as plants, including algae, as well as bacteria and fungi, possess a semipermeable cell wall that allows small molecules to pass through. Fabrega et al. [59] reported that silver NP agglomerates caused cell wall damage in algae and were able to pass through cell walls to reach the plasma membrane. There are conflicting views regarding the ability and mechanisms for cellular uptake of NMs by cells, although several studies using TEM have shown uptake. For instance, some gram-positive bacteria have been shown to take up NPs through their semipermeable membranes [60,61] and, also using TEM, AgNPs were shown to pass through the pores of the chorion in Zebrafish embryos. The TEM images in the present study (Figure 8) show cell wall breakage, plasmolysis and possible internalization of the SWCNTs following exposure to 500 µg L^−1^ SWCNTs, which is in agreement with previous reports [55]. In comparison to the cells from the control samples, the SWCNT-exposed cells showed distinct changes in the morphology of their cell walls and membranes: their size was reduced and their cell membranes deformed. These impacts could be attributable to the destructive effects of ROS, due to the fact that membrane integrity is a primary target [60,62]. Although unlikely, given their absence on control sample images, it cannot be completely ruled out that these deformations were artefacts of TEM preparation. Algal cells can be entrapped and coated by a tangled matrix of SWCNTs at concentrations ≥100 µg L^−1^. The observations in the present study elicited five interrelated effects on the algae: diminished flagellar motility, decreased light availability, decreased cell viability, growth inhibition and decreased chlorophyll *a* concentration. This chain of events could occur in the case of the release of ENMs into the aquatic environment, affecting the availability of food for filter-feeding consumers and subsequent trophic levels. Together with the observed cell wall breakage and plasmolysis, the results suggest that the SWCNTs in this study were internalized by the algae and played a role in reducing the algal viability and growth (Figure 6, Figure 7 and Figure 8, respectively), although the precise mechanism of SWCNT uptake is, at present, unknown.

### 3.2. Toxicological Response of Algae to SWCNT Exposure

Changes in chlorophyll *a* concentration have been used as an indicator of general biological response, such as algal growth rate [23]. The *T. suecica* were exposed to three replicate 500 µg L^−1^ SWCNTs. Statistically, there was no significant difference in the level of chlorophyll *a* concentration between the control and SWCNTs groups at $t_{0}$ (*p* = 0.312). However, from $t_{1}\to t_{3}$ (72 h; Figure 5), there was a significant decrease in the chlorophyll *a* concentration in the 500 µg L^−1^ SWCNT groups compared to the controls (*p* < 0.001). This observation is consistent with the significant growth inhibition and was accompanied by a significant decrease (*p* < 0.001) in cell viability (Figure 6). Whilst the concentrations represent an extreme situation and are unlikely to be encountered in a representative environmental context, they clearly demonstrate the potential impact of SWCNTs on primary production and the base of the marine food chain.

### 3.3. Response of Mussels to SWCNT Contaminated Algae

The feeding behavior of *Mytilus* sp. and other filter-feeding bivalves has important water quality implications and have consequently been thoroughly studied in both the presence and the absence of algal cells [32]. The interaction of SWCNTs with algae and the effect on the feeding behavior of mussels was observed using video-camera recordings and showed copious amounts of pseudofaecal SWCNTs expelled through the exhalant siphon of the mussel when fed algae (*T. suecica*) in the presence of SWCNTs (Figure 9). Clearly, the SWCNTs agglomerated during the experiment and affected the feeding behavior of the mussels with and without algae. Faecal material is the result of digested edible foods expelled by the exhalant siphon following passage through the digestive tract. In contrast, pseudofaeces consist of mucus-bound material rejected by the sorting mechanism of the labial palps and excreted by the exhalant siphon without being ingested. Both faecal and pseudofaecal material were observed when mussels were fed algae, SWCNTs 500 µg L^−1^ alone and SWCNTs 500 µg L^−1^ + algae (Figure 10A–E). Closer observation using light microscopy showed the different composition of pseudofaecal material when fed SWCNTs alone and SWCNTs + algae (Figure 11A,B). Selective bivalve feeding was observed using a newly developed flow cytometry technique with pseudofaeces as a proxy for feeding behavior in mussels (Figure 12). Pseudofaeces containing algal cells increased significantly (*p* = 0.008) under combined algae and SWCNT exposure, suggesting mussels largely rejected algae containing or associated with SWCNTs (Figure 10A–E). DNA damage and oxidative stress were used as ecotoxicological biomarkers of the exposure to SWCNTs in mussels fed with SWCNT-contaminated algae. A 24 h exposure to 500 µg L^−1^ SWCNTs showed significantly increased DNA strand breaks in both gill cells and haemocytes (*p* < 0.001) (Figure 13A,B) and significantly increased oxidative stress, expressed as superoxide dismutase (SOD) activity (*p* < 0.001) and lipid peroxidation in gills (*p* = 0.032) (Figure 14A,B). However, when SWCNTs were presented together with algae, the DNA damage in haemocytes and gills (*p* = 0.534; *p* = 0.998) (Figure 13A,B) and oxidative stress were not significantly increased above control levels (*p* = 0.981; *p* = 0.999) (Figure 14A,B). These results illustrate that the presence of algal cells contaminated with SWCNTs stimulated the mussels to expel copious amounts of SWCNT-containing pseudofaeces. It can therefore be concluded that the presence of algae plays a major role in mitigating the toxicity of SWCNTs to filter-feeding bivalves.

### 3.4. Trophic Transfer of SWCNTs

One of the most significant, and little understood, risks of ENMs is their potential for trophic transfer and biomagnification in food webs [63,64,65]. Trophic transfer of ENMs has been observed between planktonic crustacean grazers and secondary consumers, such as fish [65], and the possibility of human exposure has been discussed [63,64,65]. Research on the toxicity of SWCNT has primarily focused on human health [46]. However, no studies have demonstrated their indirect impact on humans through the food chain. The risk assessment of NMs or NPs is not only restricted to their concentration in the environment and their toxicity to organisms [66,67], but also depends on their bioaccumulation and biomagnification in aquatic food chains [15,16,37,38,39,40,41,42,43,44,45,46,47,48,49,50,51,52,53,54,55,56,57,58,59,60,61,62,63,64,65], although the ecological impacts of this are mostly unknown. Together with the lifecycle of CNTs in the environment, these key and emerging knowledge gaps are critical obstacles in understanding the physical interactions between algae and mussels, and the trophic transfer of SWCNTs through the food web [68,69]. In the present study, the trophic transfer of SWCNT from algae to mussels was assessed by exposing algae to SWCNT 500 µg L^−1^ for seven days, and then feeding the exposed algal cells to mussels for 10 min (the clearance time in the experimental setup). Histological analysis and Raman microspectroscopy showed and confirmed algal cells containing SWCNTs in the mussel gut epithelium (Figure 15B–D).

## 4. Conclusions

SWCNTs were found to significantly affect the viability of the algae *T. suecica* and we conclude that (i) mussels continue to filter on algae despite the presence of SWCNTs; (ii) although mussels are able to selectively exclude large numbers of SWCNT-contaminated algae, some are nevertheless ingested; (iii) mussels exposed to SWCNTs produced large amounts of pseudofaeces containing SWCNTs-contaminated algae; (iv) SWCNTs can be transferred from primary producers to filter-feeding consumers; and (v) the transferred SWCNTs can elicit a measurable toxic response. Further research is required to understand the process of uptake and toxic kinetics of SWCNTs in mussels.

## Figures and Tables

**Figure 1 nanomaterials-12-04363-f001:**
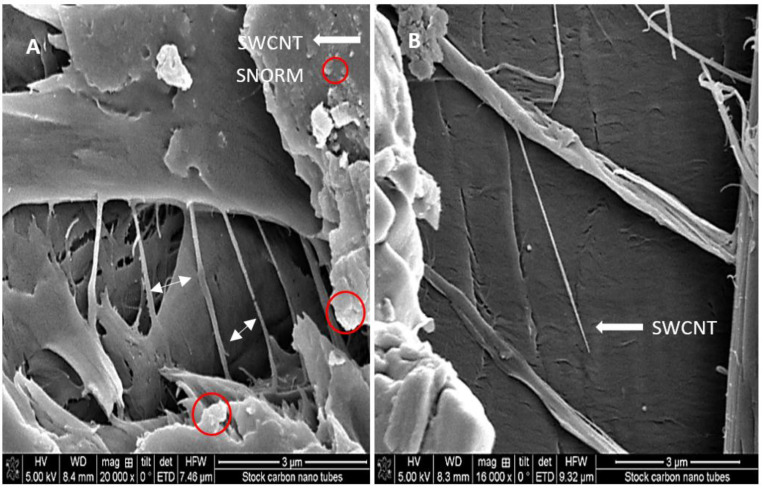
Scanning electron microscope images of SWCNT. (**A**) Crystallized SWNT-SRNOM films (**B**) SEM image of an individual SWCNT partly encased in SRNOM.

**Figure 2 nanomaterials-12-04363-f002:**
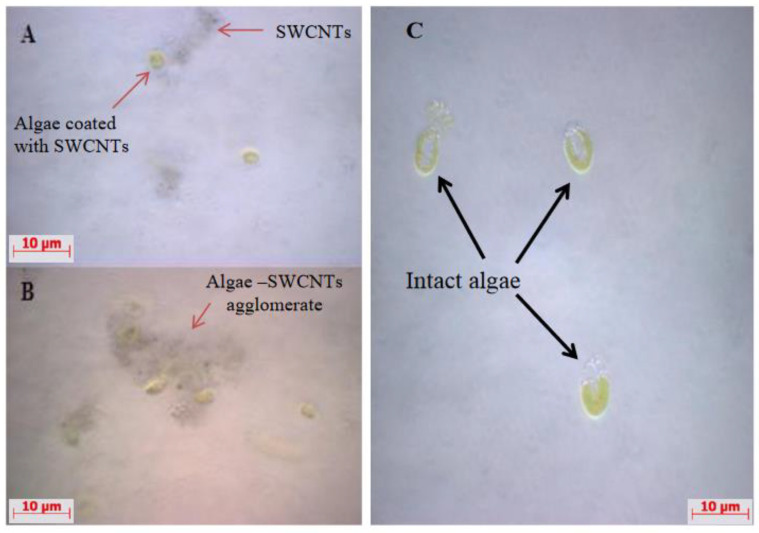
*Tetraselmis suecica* in the presence and absence of SWCNTs; light micrographs for SWCNT-exposed (**A**) 100 µg L^−1^, (**B**) 500 µg L^−1^, and Control (**C**). Images show a dark coloration on the surface of treated algae (**A**,**B**) compared to the clean surface in the control algae (**C**).

**Figure 3 nanomaterials-12-04363-f003:**
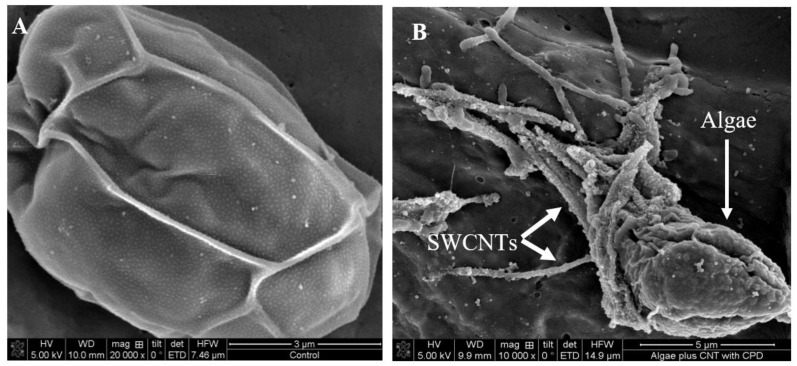
SEM images of *T. suecica* from control samples (**A**) and from culture medium containing final 500 µg L^−1^ SWCNTs (**B**); algae (appear surrounded by SWCNTs agglomerates).

**Figure 4 nanomaterials-12-04363-f004:**
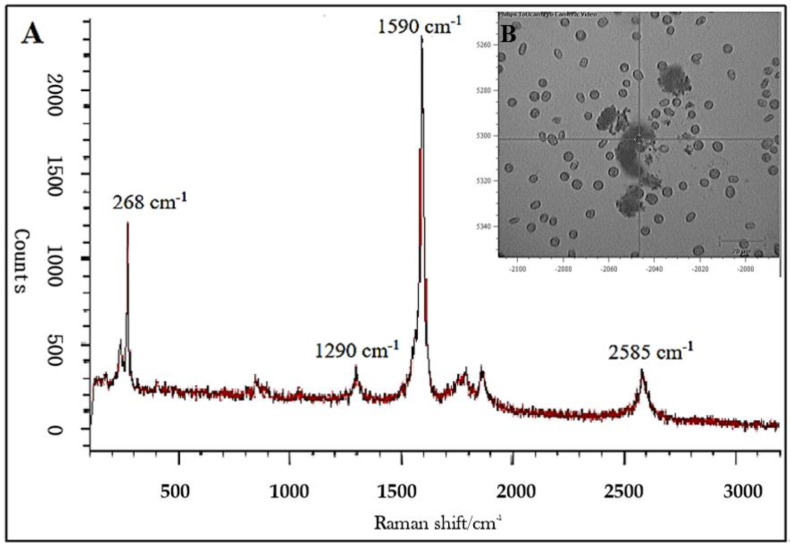
A. Raman spectrum of Algal-SWCNT interaction, (**A**) Magnification (×20) of algal cells exposed to 500 µg L^−1^ single-walled carbon nanotubes (SWCNTs) for 24 h. The white circle in the center of (**B**) marks the position of the Raman excitation beam spot, which was focused to a diameter of 1.7 µm. A representative Raman spectrum was acquired from SWCNT stock, algae cells spiked with 500 µg L^−1^ SWCNT for 24 h (**A**). The spectrum was collected using a 50×, 0.75 numerical aperture microscope objective lens. The spectrum clearly shows the characteristic of (red peak) SWCNTs stock and (black peak) algal cells exposed to 500 µg L^−1^ SWCNTs: radial breathing mode at 268 cm^−1^, D band at 1290 cm^−1^, G band at 1590 cm^−1^, and G’ band at 2585 cm^−1^.

**Figure 5 nanomaterials-12-04363-f005:**
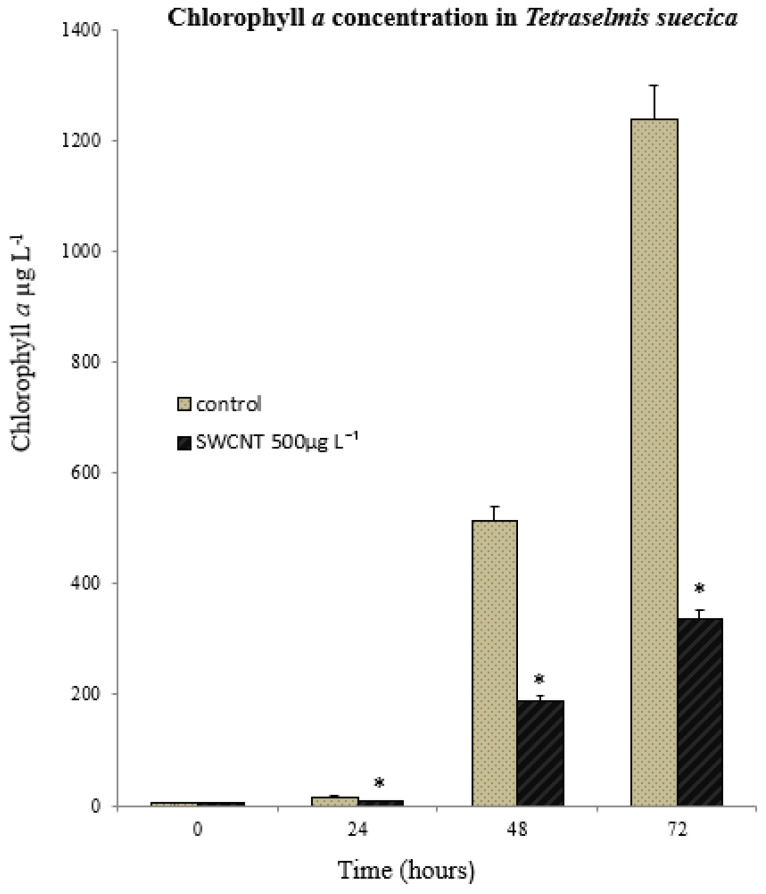
*Tetraselmis suecica* growth (Chlorophyll *a*). (*) indicates, significantly decreased chlorophyll *a* concentrations compared to the control (One Way ANOVA: F = 1.0512, df_num_ = 1, means ± SD; n = 3; *p* < 0.001).

**Figure 6 nanomaterials-12-04363-f006:**
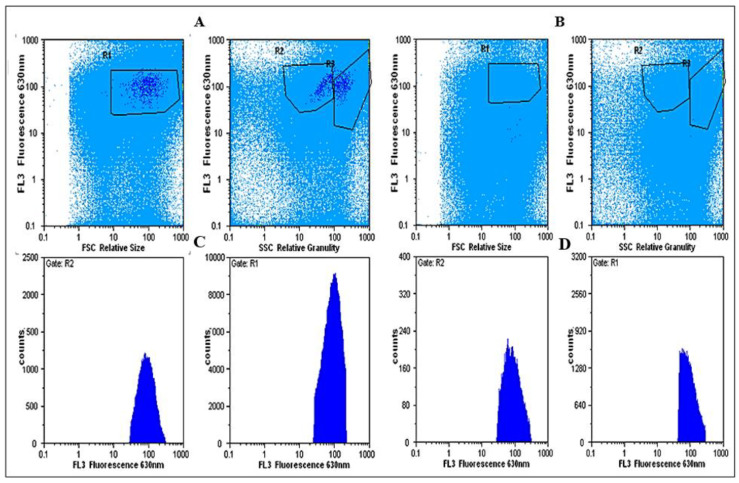
Cell viability. Peaks (**A**,**C**) control and (**B**,**D**) 500 µg L^−1^ SWCNT show the viable *T. suecica* cells using flow cytometry (Cyflow). (**A**,**B**) R1 is the defined cell population from which other graphs are extrapolated, and R1 represents the cell population as a histograms. (**C**,**D**) Shows the ratio of live and dead cells. Live cells appear as one large fluorescence peak, whereas dead or necrotic cells are represented as one small fluorescence peak. Statistically, there was a significant decrease in cell viability of algal cells when exposed to 500 µg L^−1^ SWCNT (One Way ANOVA: F = 1.91, df_num_ = 1, means ± SD; n = 3; *p* < 0.001).

**Figure 7 nanomaterials-12-04363-f007:**
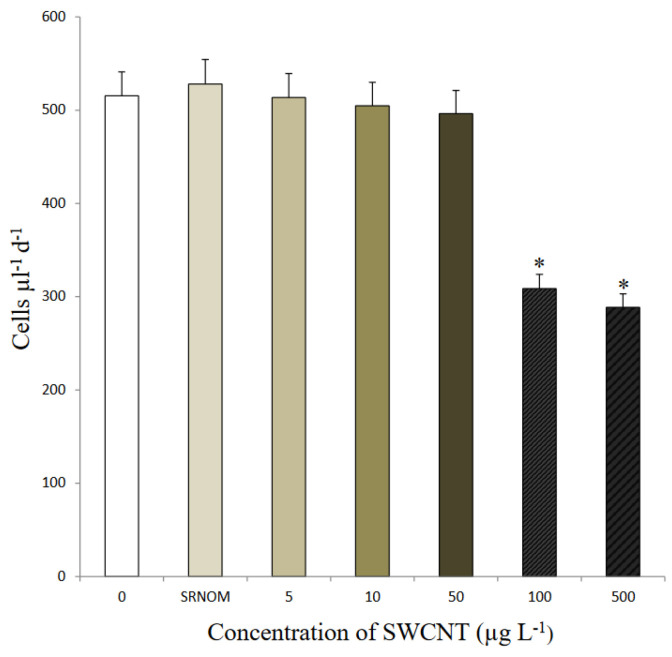
*Tetraselmis suecica* growth rate during exposure to SWCNTs prepared in 0.02% SRNOM at nominal concentrations (5 µg L^−1^, 10 µg L^−1^, 50 µg L^−1^, 100 µg L^−1^, and 500 µg L^−1^). Statistically, there was no significant difference between SRNOM, 5 µg L^−1^, 10 µg L^−1^, 50 µg L^−1^ and control groups; (*) significant growth inhibition occurred >100 µg L^−1^ (One Way ANOVA: F = 1.82 df_num_ = 5, means ± SD; n = 3; *p* < 0.001).

**Figure 8 nanomaterials-12-04363-f008:**
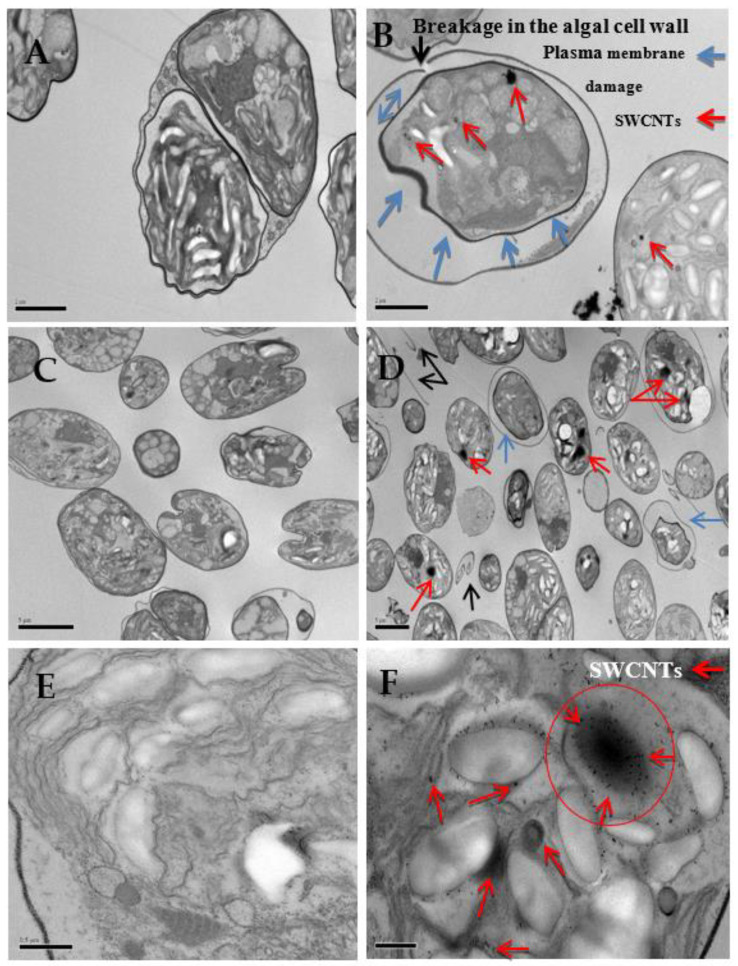
TEM images of control cells with intact algal cell wall and plasma membrane (**A**,**C**,**E**) and from culture medium containing final 500 µg L^−1^ SWCNTs (**B**,**D**,**F**); cell wall breakage (black arrows); plasmolysis (blue arrows) and internalization of the SWCNTs (red arrows). Scale bars 2 μm (**A**,**B**); 5 μm (**C**,**D**); 0.5 μm (**E**,**F**).

**Figure 9 nanomaterials-12-04363-f009:**
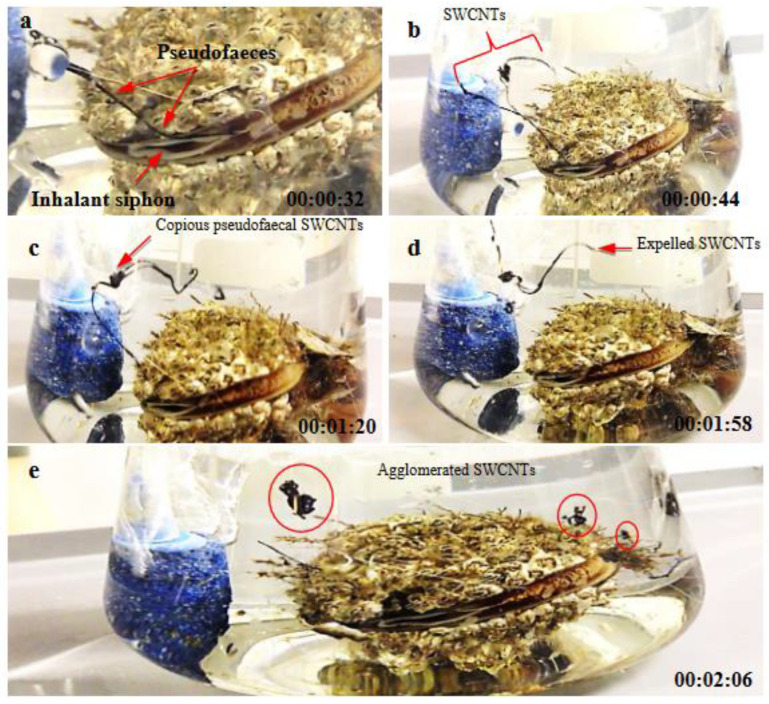
Feeding behavior of mussels when fed SWCNTs + algae: (**a**) mussel starts to expel SWCNTs; (**b**,**c**) long black nanotubes still attached to the inhalant siphon of mussel; and (**d**,**e**) SWCNTs have become agglomerated or aggregated in seawater.

**Figure 10 nanomaterials-12-04363-f010:**
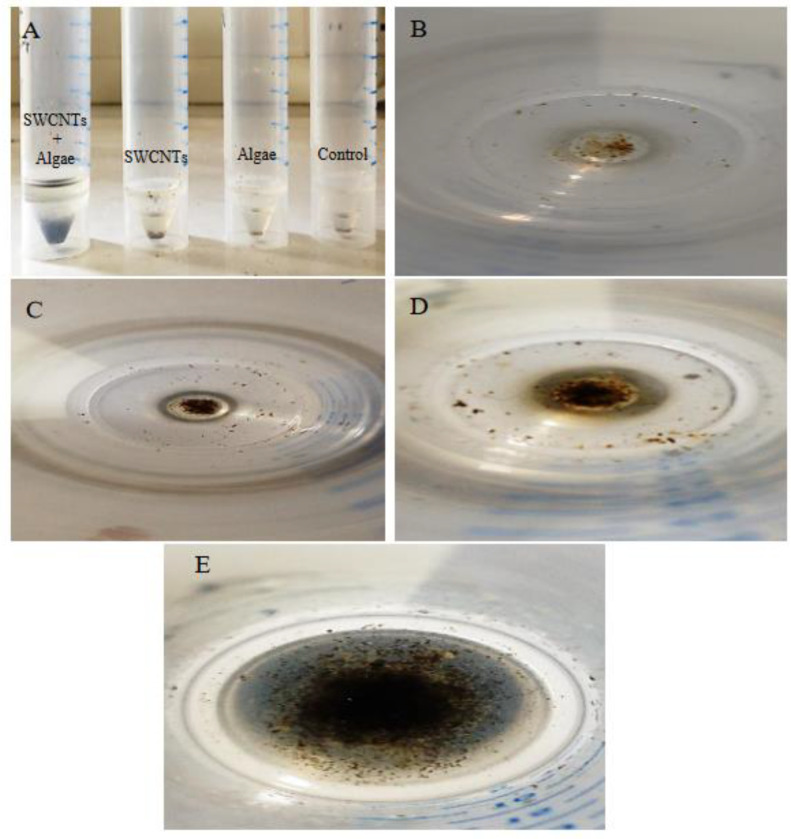
Feeding behavior of the mussel *Mytilus edulis*. (**A**) Faecal and pseudofaecal material expelled by the exhalant /inhalant siphons of the mussels when fed the algae, SWCNTs 500 µg L^−1^ alone and SWCNTs 500 µg L^−1^ + algae (lateral view of tubes). (**B**,**C**) Faecal material expelled by the exhalant siphon of the mussel when fed the *Tetraselmis suecica* alone. (**D**) Pseudofaecal material expelled by the inhalant siphon of the mussels when fed the SWCNTs 500 µg L^−1^ alone and (**E**) in combination with algae ((**B**–**E**): vertical view).

**Figure 11 nanomaterials-12-04363-f011:**
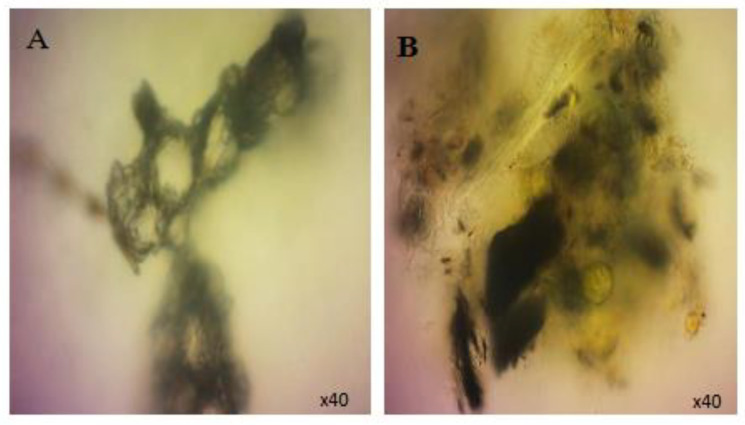
Light micrographs of faecal and pseudofaecal algal cells. (**A**) shows Pseudofaecal material expelled by the exhalant siphon of the mussels when fed the SWCNTs 500 µgL^−1^ alone, while figures (**B**) shows pseudofaecal algal cells and SWCNTs expelled by the exhalant siphon of the mussel when fed SWCNTs and *Tetraselmis suecica*.

**Figure 12 nanomaterials-12-04363-f012:**
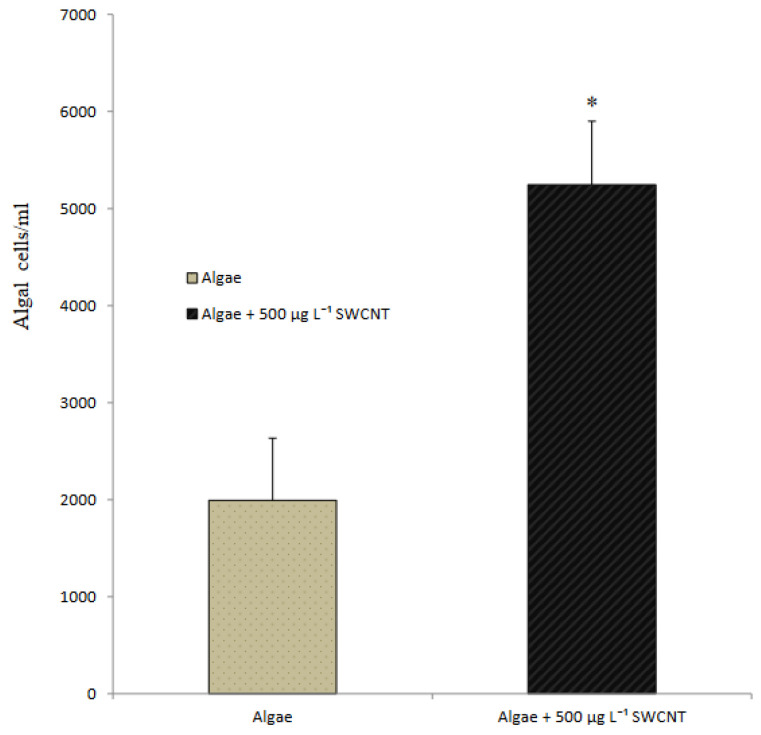
Mean (±SD) pseudofaeces production of mussels fed algae and 500 µg L^−1^ SWCNTs alone, and in combination; * significantly different in pseudofaeces production of mussels fed algae and algae + 500 µg L^−1^ SWCNTs (One Way ANOVA: F = 2.5217, df_num_ = 1, means ± SD; n = 3; *p* = 0.008).

**Figure 13 nanomaterials-12-04363-f013:**
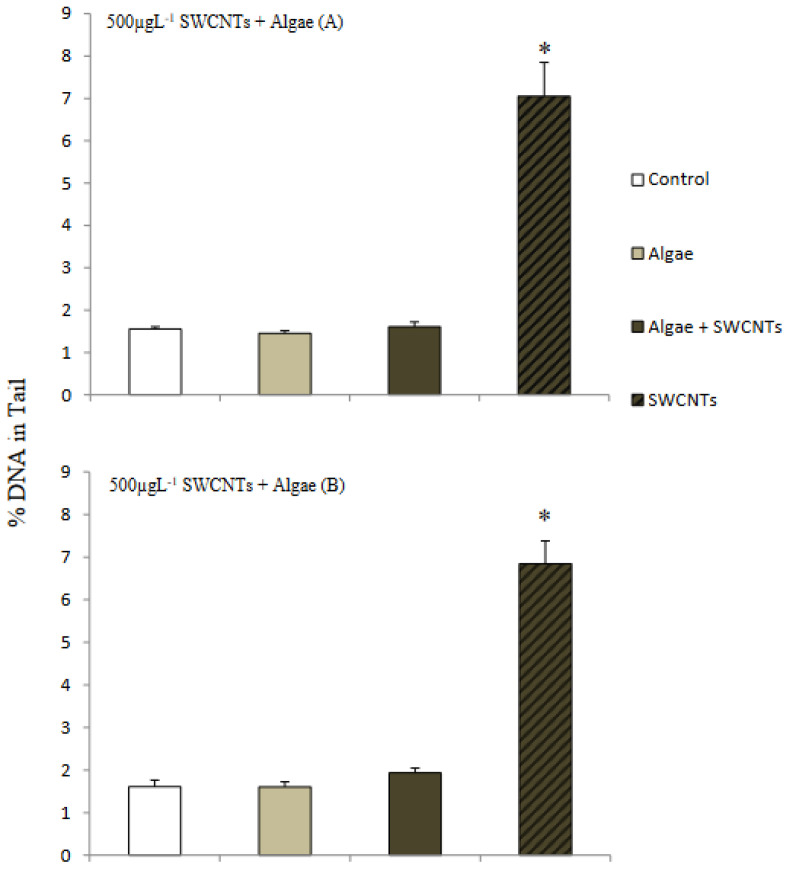
Mean (±SD) DNA strand breaks (Comet assay) in mussel (**A**) haemocytes and (**B**) gills after feeding on algae (*Tetraselmis suecica*), SWCNT 500 µg L^−1^ alone and *Tetraselmis suecica* + SWCNTs 500 µg L^−1^ for 24 h. * significantly different from control, algae and algae + SWCNTs 500 µg L^−1^ (One Way ANOVA: F = 1.83, df_num_ = 3, means ± SD; n = 3; *p* < 0.001).

**Figure 14 nanomaterials-12-04363-f014:**
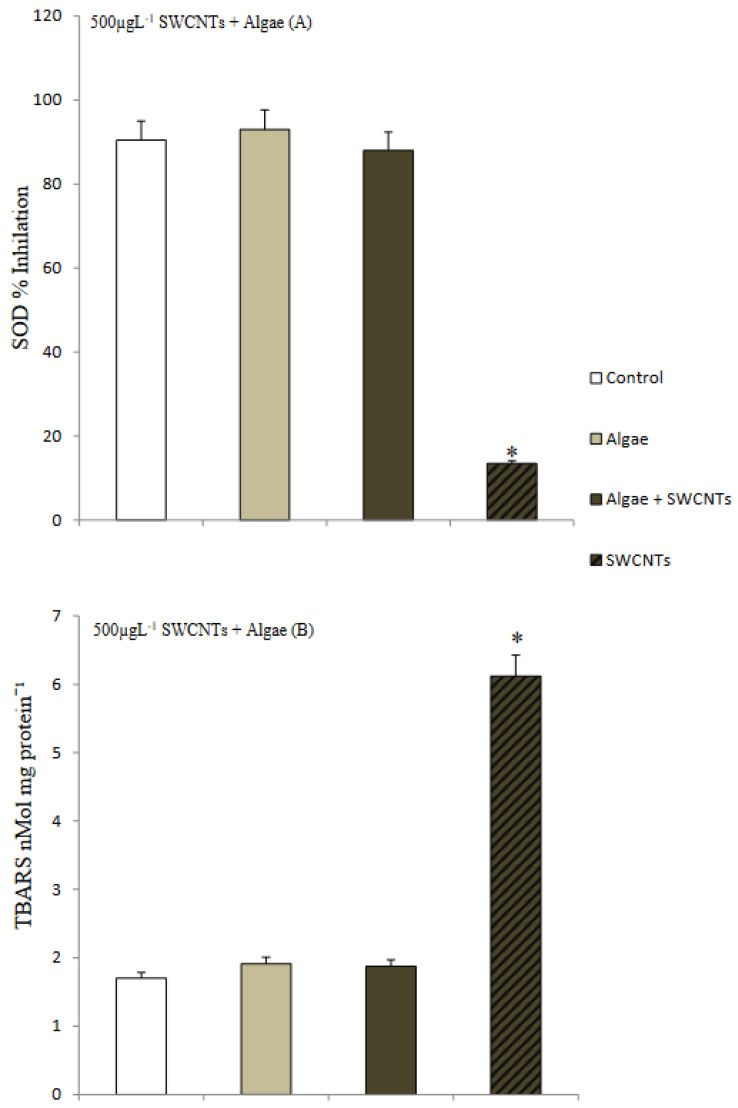
Oxidative stress in mussel gill tissue after feeding on algae (*Tetraselmis suecica*), SWCNT 500 µg L^−1^ alone and *Tetraselmis suecica* + SWCNTs 500 µg L^−1^ for 24 h. (**A**) Superoxide dismutase (SOD) activity (One Way ANOVA: F = 6.7, df_num_ = 3, means ± SD; n = 3; *p* < 0.001); (**B**) Lipidperoxidation (TBARS); * significantly different from control, algae and algae + SWCNTs 500 µg L^−1^ (One Way ANOVA: F = 5.8 df_num_ = 3, means ± SD; n = 3; *p* < 0.001).

**Figure 15 nanomaterials-12-04363-f015:**
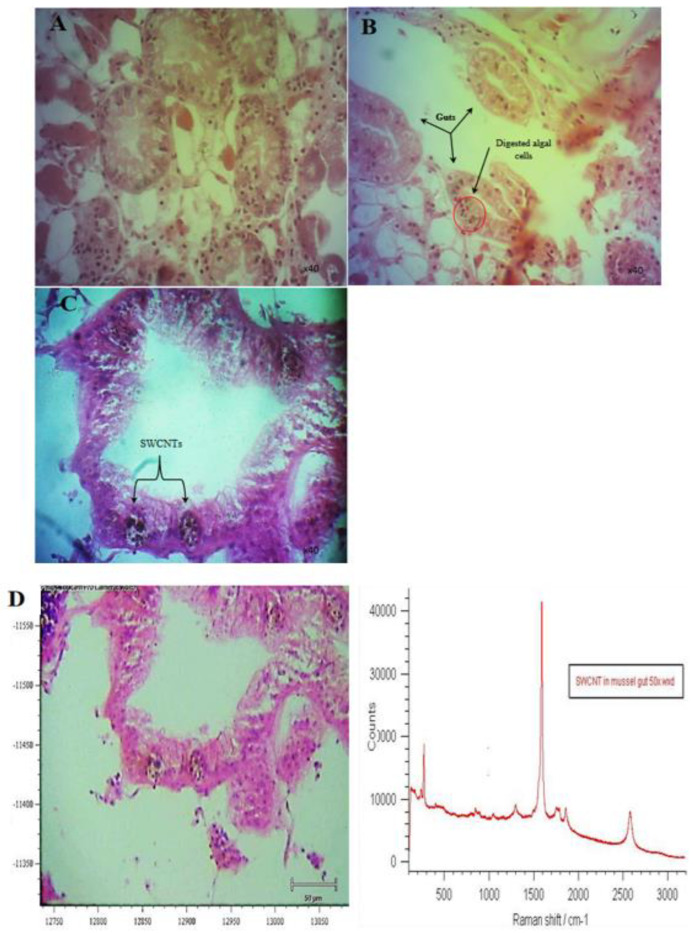
The gut from mussels exposed to algal cells-SWCNT 500 µg L^−1^. Histological sections of mussel’s gut in (**A**) control tissue, (**B**) digested algal cells, and (**C**) SWCNTs in gut, (**D**) Raman spectrum of the histological sections of the digestive gland of mussel confirming uptake of SWCNTs; scale bar (50 µm).

**Table 1 nanomaterials-12-04363-t001:** Single-walled carbon nanotube (SWCNT) dispersed in SRNOM: concentration dependent agglomerate size characterization.

SWCNT (µg L^−1^)	Zeta Potential	DLS (nm)
5	−8.84	475
10	−10.83	1384
50	−10.13	1740
100	−15.93	4982
500	−13.73	6206

pH 8.4; salinity 32 (±1) ppt; DLS: Dynamic Light Scattering; SRNOM: Suwannee River Nature Organic Matter.

## Data Availability

The data presented in this study are available on request from the corresponding author.

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
