# Peer review of "Trophic Transfer of Single-Walled Carbon Nanotubes at the Base of the Food Chain and Toxicological Response"

_nanomaterials, 2022, doi:10.3390/nano12244363_

Round 1
Reviewer 1 Report
Some small corrections are indicated in the manuscript

Author Response
Dear Reviewer, Response to Reviewer 1 Comments: Point 1: I found that in the abstract and as per your valuable comments it has been changed to (Mussels fed with algae) Point 2: In section 2.3, as per your valuable comments, the sentence was amended to (The highest concentration was used...etc).

Reviewer 2 Report
Authors conducted interesting experimental studies on CNT's toxicity propagated along the food chain. They studied the CNT-algae interaction and the CNT-containing algae fed to mussel. Overall, this is an interesting study that raises the concern about the safety of CNT. I recommend its publication after authors improve the quality of figures (especially #4 and #6).
Author Response
Response to Reviewer 2 Comments, Many thanks for your valuable comments.
Point 1: The quality of figures were improved (especially #4 and #6).

Reviewer 3 Report
The paper is well structured and writen. The experimental design is experimentally sound and the objectives clearly formulated.
As for specific aspects it should be noted that in section 2.12.4 Data Analysis the software package used to conduct the analysis should be included as well as its citation in a suitable format. Furthermore, in 3. Results & Discussion section whenever statistical results are referred to (either significant or non significant) the statistical method used, the value of the test, its degrees of freedom and p-value should be presented (e.g. ANOVA: F=2.71, dfnum=3, dfden=27, p=0.02) as presenting the p-value is not enough to the reader to assess the validity of the statistical results.
Author Response
Response to Reviewer 1 Comments:
Dear Reviewer,
Thank you for your valuable comments. We added the F and df under and p value in each figure as your valuable requested.
Kind Regards
Majed
